# Nutritional, Bioactive, and Flavor Components of Giant Stropharia (*Stropharia rugoso-annulata*): A Review

**DOI:** 10.3390/jof9080792

**Published:** 2023-07-28

**Authors:** Lei Huang, Chunmei He, Can Si, Hongyu Shi, Jun Duan

**Affiliations:** 1Key Laboratory of South China Agricultural Plant Molecular Analysis and Gene Improvement, South China Botanical Garden, Chinese Academy of Sciences, Guangzhou 510650, China; huanglei2020@scbg.ac.cn (L.H.); hechunmei2012@scbg.ac.cn (C.H.); cansi2016@scbg.ac.cn (C.S.); shihongyu@scbg.ac.cn (H.S.); 2University of Chinese Academy of Sciences, Beijing 100049, China

**Keywords:** *S. rugoso-annulata*, nutritional components, bioactive components, flavor components, drying

## Abstract

Giant Stropharia (*S. rugoso-annulata*) is an edible mushroom recommended for consumption by the Food and Agriculture Organization of the United Nations. It possesses significant culinary and medicinal functionalities. The characteristics of this mushroom include high protein content, abundant bioactive compounds, delicious and sweet taste, and pleasant aroma. In recent years, the *S. rugoso-annulata* industry has seen strong growth, especially in China. This article presents the first comprehensive and systematic review of the nutritional, bioactive, and flavor components of *S. rugoso-annulata*, as well as their influencing factors. This article provides scientific evidence for the production of high-quality *S. rugoso-annulata* mushrooms, the extraction of bioactive components, post-harvest storage, and culinary processing, aiming to promote the consumption of *S. rugoso-annulata* and the health of consumers.

## 1. Introduction

*Stropharia rugoso-annulata*, also known as Giant Stropharia, Wine-cap mushroom, is a renowned edible fungus recommended for consumption by the Food and Agriculture Organization of the United Nations due to its potential to alleviate various illnesses associated with cancer [1]. Not only is it a high-protein, low-fat food rich in minerals, vitamins, and dietary fiber [2], but it is also a quality raw material for functional components such as fungal polysaccharides [3], sterols [4], and taste peptides [5], and can also be processed into various flavorful foods [6,7]. With particularly simple cultivation techniques and high yield [8], it holds significant application value in various fields such as dietary nutrition supply, bioactive compound extraction, functional food production, and flavorful food processing.

*S. rugoso-annulata* was first domesticated in Germany in 1969. Subsequently, it was introduced for cultivation in other European countries and the United States [8]. In 1989, the yearly production of *S. rugoso-annulata* reached approximately 1300 tons in Europe [9]. China imported a strain of *S. rugoso-annulata* from Poland in the 1980s and began widespread cultivation in the 1990s [10]. In recent years, *S. rugoso-annulata* has been rapidly promoted and widely cultivated throughout China. In 2021, China′s fresh *S. rugoso-annulata* mushroom production exceeded 210,000 tons, representing a surge of 43% compared to 2019 [11].

In recent years, numerous research has been reported on the food properties of *S. rugoso-annulata*, particularly its flavor components. Jing et al. reported on the nutritional components, some bioactive substances, and heavy metal content of *S. rugoso-annulata* [2]; Chen et al. reported on the flavor components of both fresh and dried *S. rugoso-annulata* [12]; Li et al. reported on the flavor substances produced by the liquid fermentation of *S. rugoso-annulata* mycelium [13]; and Lu et al. reported the flavor components of *S. rugoso-annulata* soup processed by different methods [14,15].

This article provides a comprehensive review of the research progress on the nutritional, bioactive, and flavor components of *S. rugoso-annulata*, as well as their influencing factors. It is a systematic theoretical reference for the production of high-quality *S. rugoso-annulata* mushrooms, the extraction of bioactive ingredients, post-harvest storage, and culinary processing. It can promote the consumption of *S. rugoso-annulata* and contribute to the health of consumers.

## 2. Nutritional Components of *S. rugoso-annulata*

As a food item, fresh mushrooms of *S. rugoso-annulata* have a high moisture content, with an average moisture content of 92% and an average dry matter content of 8% [8]. The subsequent content data are reported on a dry matter basis unless otherwise specified. The basic nutritional components of *S. rugoso-annulata* (dried by hot air drying, HAD) are ranked in order from highest to lowest content as follows: carbohydrates (45.17–54.60%), protein (25.75–34.17%), minerals (ash, 7.40–9.62%), dietary fiber (5.25–7.99%), fat (1.33–2.30%), and vitamins (<0.33%) [2,16,17]. *S. rugoso-annulata* contains 18 out of the 20 protein-composing amino acids, excluding Asn and Gln. It is rich in all eight essential amino acids for the human body including Leu, Ile, Val, Phe, Met, Trp, Thr, and Lys [2]. The total content of amino acids in this mushroom ranges from 18.89% to 31.01%, with essential amino acids accounting for 6.54% to 11.70%, and non-essential amino acids accounting for 7.19% to 19.97% [2,16,18,19]. According to most studies, Glu has the highest content (2.88–6.84%), followed by Asp (1.72–3.07%) [16,18,19,20]. However, some studies have suggested that Ile has the highest content (6.32%), followed by Tyr (1.86%) and Glu (1.82%) [2]. This difference may be related to different regions or the substrates used for cultivation.

*S. rugoso-annulata* mushrooms contain a variety of B-complex vitamins, notably riboflavin (B2), niacin (B3), pantothenic acid (B5), pyridoxine (B6), folic acid (B9), and cobalamin (B12) [2]. Additionally, they possess vitamin C [21], as well as ergosterol, a precursor to vitamin D2 [17]. The ergosterol content is abundant, up to 0.23% [17]. The contents of vitamin C (0.53‰) and B3 (0.39‰) are also relatively high [2,21].

*S. rugoso-annulata* mushrooms contain a variety of essential mineral elements, including K, P, Ca, Mn, Cu, Na, Fe, Zn, Mg, and Se, and can also accumulate heavy metal elements such as As, Cd, Hg, Pb, and Cr from the soil [2,16,21]. Among them, the content of necessary macro elements such as Na, K, Mg, P, and Ca exceeds 0.1%, with K having the highest amount (2.68–3.48%) [2,16] and P as the second highest (0.82–0.87%) [2,21].

*S. rugoso-annulata* accumulates Zn and Se from cultivation medium or substrate, serving as an excellent carrier of organic Zn and Se to the human body. The mycelium of *S. rugoso-annulata* cultured in a Zn-Se-rich liquid medium can produce organic Zn of up to 0.21 mg/g and organic Se of up to 0.82 mg/g [22]. The fruiting body of *S. rugoso-annulata* grown in the wild can contain up to 2.57 mg/kg of Se [2]. The addition of Se as fertilizer to cultivation substrates can increase the Se content of the *S. rugoso-annulata* fruiting body to 3.93 mg/kg [23].

The mycelium of *S. rugoso-annulata* exhibits a fatty acid content ranging from 3.64 to 7.52 mg/g, with unsaturated fatty acids constituting over 77% of the total fatty acids. The predominant unsaturated fatty acid is linoleic acid (C18:2), a polyunsaturated fatty acid, while the main saturated fatty acid is palmitic acid (C16:0). Linoleic acid has the highest content, comprising over 57% of the total fatty acids, followed by palmitic acid, accounting for over 13% of the total fatty acids [24]. Other fatty acids such as oleic acid (C18:1), palmitoleic acid (C16:1), stearic acid (C18:0), and lignoceric acid (C24:0) also have relatively high contents [24,25].

### 2.1. Nutritional Components of S. rugoso-annulata Processed by Different Drying Methods

Dried *S. rugoso-annulata* mushrooms are commonly produced using hot air drying (HAD) at temperatures ranging from 45 °C to 55 °C [12,26,27]. In addition to HAD, there are also other methods for drying such as vacuum freeze-drying (VFD), natural air drying (ND, NAD), microwave drying (MWD), and microwave vacuum drying (MVD) [27,28,29,30,31]. The dry matter and crude fat content of *S. rugoso-annulata* followed the order of VFD > HAD > ND, while the protein and polysaccharide content followed the order of VFD > ND > HAD. The differences between the three drying methods are significant. *S. rugoso-annulata* produced by VFD had a significantly higher content of protein (25.09%), polysaccharides (16.03%), and fat (6.05%) than those produced by HAD and ND. Additionally, VFD retained the fresh mushroom’s appearance and color effectively [29]. VFD is suitable for the production of high-quality *S. rugoso-annulata* mushrooms, while the method of ND followed by HAD is suitable for the production of regular products.

### 2.2. Nutritional Components of S. rugoso-annulata at Different Developmental Stages

As shown in Figure 1, the developmental process of the *S. rugoso-annulata* fruiting body can be divided into five stages, namely, S1 (the button stage), S2 (the gill formation stage), S3 (the early maturity stage), S4 (the maturity stage), and S5 (the parachute stage), in chronological order. The ash content reaches the highest level at the button stage and decreases significantly afterward. The total carbohydrate content increases continuously throughout the entire developmental process. The crude protein content remains relatively stable but declines during the parachute stage. The crude fiber and crude fat content did not vary much at each stage [32,33]. The crude protein content of *S. rugoso-annulata* was significantly higher in the unopened stage than in the opened stage [34]. S4 and S2 are the stages when *S. rugoso-annulata* exhibits a higher overall nutritional quality, providing a reference for the timely harvesting of *S. rugoso-annulata* [32].

### 2.3. Nutritional Components of Different Parts of S. rugoso-annulata

As shown in Table 1, the nutrient components of both the pileus and the stipe of *S. rugoso-annulata* are the same, but their contents are different. The pileus presents a significantly higher content of crude protein, total amino acids, crude fat, ash, Mn, and Zn than the stipe, resulting in a higher nutritional value [35].

## 3. Bioactive Components of *S. rugoso-annulata*

### 3.1. Soluble Polysaccharides

Soluble polysaccharides are the main bioactive components of *S. rugoso-annulata* [37]. They exhibit beneficial health properties like antioxidant [26,37,38,39], anti-tumor [40], anti-inflammatory [41], antimicrobial [37], immunomodulatory [42], hypoglycemic [43], hepatoprotective [44], mitochondrial swelling-inhibiting, and erythrocyte hemolysis-inhibiting activities [45], as well as plant growth promotion activity [46,47].

*S. rugoso-annulata* soluble polysaccharides are heteropolysaccharides, with glucose and galactose being the most common composing monosaccharides (Table 2) [3,28,41,48,49]. They may also contain mannose, arabinose [28], fucose [41], glucuronic acid, ribose [49], fructose, xylose, and rhamnose [3]. Glucose is the most abundant monosaccharide in the soluble polysaccharides of the fruiting body [3,28,41,48,49]. Based on the presence or absence of acidic groups such as glucuronic acid or sulfate groups in the composition of monosaccharides, *S. rugoso-annulata* polysaccharides can be divided into neutral polysaccharides and acidic polysaccharides [26,28,49]. Neutral polysaccharides mainly have a (1→6)-α-d-glucan or (1→6)-α-d-galactan as the main chain, while acidic polysaccharides mainly have a (1→6)-α-d-glucan or (1→3)-β-d-glucan as the main chain [28,49]. Both types of polysaccharides have α- and β-glycosidic bonds and a molecular weight of 13.28–8992 kDa [26,42]. The polysaccharides of *S. rugoso-annulata* mycelium are divided into intracellular polysaccharides and extracellular polysaccharides, composed of pyranose and connected by β-glycosidic bonds [40,48,50]. The content of soluble polysaccharides in the fruiting body ranges from 6.98% to 13.25% [37,49,51], while that in the mycelium ranges from 15.12% to 22.37% [50,52]. Studies have shown that appropriate chemical modifications (carboxymethylation, phosphorylation, and acetylation) can improve the physicochemical properties and biological activities of *S. rugoso-annulata* polysaccharides [41,44]. For example, carboxymethylated *S. rugoso-annulata* polysaccharides exhibit stable DPPH scavenging and reducing abilities, α-glucosidase inhibitory activity, and α-amylase inhibitory activity; phosphorylated *S. rugoso-annulata* polysaccharides have significant hydroxyl radical scavenging ability [41]; and acetylated *S. rugoso-annulata* polysaccharides have activity in alleviating non-alcoholic fatty liver disease and inhibiting fat synthesis [44]. No significant correlation was found between the antioxidant property and the molecular weight of the crude *S. rugoso-annulata* polysaccharides. Different extraction methods have no significant effect on the chemical structure of the crude polysaccharides [26]. Acidic polysaccharides exhibit stronger antioxidant activity than neutral polysaccharides [28,49].

Drying methods can affect the proportion of monosaccharides as well as the antioxidant activity of *S. rugoso-annulata* polysaccharides. The polysaccharides obtained through HAD have a higher proportion of glucose and a lower proportion of galactose compared to polysaccharides obtained through FVD, and they also exhibit stronger antioxidant activity [28].

Soluble polysaccharides of *S. rugoso-annulata* are traditionally extracted by the water extraction–alcohol precipitation method. Ultrasound or microwave assistance is often employed during water extraction to increase the yield and shorten the extraction time [26,28,37,49]. Recently, a new method, three-phase extraction, has been reported for the extraction of *S. rugoso-annulata* polysaccharides. This method utilizes a tert-butanol-(NH_4_)_2_SO_4_-water system, which avoids the need for an alcohol precipitation process and achieves the separation of polysaccharides faster [3]. The highest yield from the *S. rugoso-annulata* fruiting body is achieved by an ultrasound-assisted extraction process (62 °C, 1:30 *w*/*v*, ultrasound 62 min), with a polysaccharide yield of 13.25% [37]. The highest yield from the *S. rugoso-annulata* mycelium is achieved by an ultrasound-assisted extraction process (63.1 °C, 1:15 *w*/*v*, ultrasound 16.33 min), with a polysaccharide yield of 22.37% [52].

### 3.2. Other Bioactive Components

In addition to polysaccharides, *S. rugoso-annulata* contains many other bioactive components including protein [54], phenols [55], triterpenes [2], flavonoids [56], sterols [4,57,58], lectin [59], oligopeptides [60], and nucleosides [61]. The optimal extraction processes for these components are shown in Table 3.

Triterpenes have anti-tumor activity [62], and the content of triterpene in *S. rugoso-annulata* is 1.42% [2]. *S. rugoso-annulata* contains a variety of sterols, including ergosterol, seven sterols with unique carbon skeletons, four sterols named strophasterol A, B, C, and D, and two sterols with a unique ether ring structure (C_28_H_44_O_4_ and C_30_H_50_O_3_) [57,58,63,64,65]. Sterols in *S. rugoso-annulata* have bioactivities such as inhibiting osteoclast formation, inhibiting fungi growth, weakening endoplasmic reticulum stress to protect neuronal cells, and inhibiting toxic carotenoids [4,57,58]. *S. rugoso-annulata* also contains lectin (SRL), a 38 kD molecule of protein with a unique N-terminal sequence, which has activity in inhibiting the proliferation of Hep G2 liver cancer cells, L1210 leukemia cells, and HIV-1 reverse transcriptase [59]. Oligopeptides (octapeptides, nonapeptides, and decapeptides) in *S. rugoso-annulata* have activity in inhibiting angiotensin-converting enzymes [60]. *S. rugoso-annulata*’s petroleum ether extract and chloroform extract have anti-fatigue activity, with the common components being ergosterol and 5α-ergosta-7, 22-dien-3β-ol [65]. *S. rugoso-annulata*’s ethyl acetate extract has strong antioxidant activity, with nucleosides as the main active component [61].

**Table 3 jof-09-00792-t003:** Bioactivity and optimal extraction conditions of other bioactive components in the *S. rugoso-annulata* fruiting body.

Components	Bioactivity	Optimal Extraction Conditions	Yield	References
Protein	Antioxidant activity and scavenging ability on DPPH radicals and hydroxyl radicals	Distilled water, pH = 12, 1:30 (*w*/*v*), 60 min, 45 °C	37.54%	[54]
Oligopeptide	Antioxidant, ACE inhibitory activity	Pure water, 1:20 (*w*/*v*), ultrasound 120 w–400 w, 20 kHz, 10–35 min	11.04–23.02%	[60]
Ergosterol	Precursor of Vitamin D2, anti-cancer, anti-aging	Ethanol 100%, 1:30 (*w*/*v*), 30 min, ultrasound, 3 repeats	0.23%	[17]
Phenol	Antioxidant, antibacterial, anticancer, anti-aging, and inhibition of cholesterol elevation	Ethanol 30%, 1:20 (*w*/*v*), 70 °C, 1 h	11.00%	[55]
		Ethanol 30%, 1:20 (*w*/*v*), 60 °C, 6 min, ultrasound 240 w	6.71%	
		Ethanol 35%, 1:20 (*w*/*v*), 2.5 min, microwave 640 w	5.32%	
Polyphenol	Antioxidation, antivirus, antibacterial	Ethanol 64.68%, 53.1 °C, ultrasound 39.3 min	1.66%	[66]
Flavonoid	Antivirus, antioxidant, antibacterial, and protection of the cardiovascular and cerebrovascular systems.	Ethanol 30%, 1:15 (*w*/*v*), ultrasound 120 w, 1 h	1.14%	[56]

## 4. Flavor Components of *S. rugoso-annulata*

The flavor of edible mushrooms includes aroma and taste. The aroma is mainly produced by volatile compounds that are perceived by the nose, while the taste is produced by non-volatile compounds that are perceived by the tongue. The intensity of the flavor is determined by the concentration of flavor compounds and their perception thresholds [67]. The taste activity value (TAV) and odor activity value (OAV) are commonly used to evaluate the non-volatile or volatile compounds in edible mushrooms. TAV or OAV is the ratio of the concentration of a certain non-volatile or volatile compound to its taste or odor threshold concentration, reflecting the contribution of a single compound to the overall taste or odor. TAV or OAV > 1 indicates that the compound contributes to the taste or odor, and the larger the value, the greater the contribution [6,12]. The equivalent umami concentration (EUC) is often used to evaluate the umami taste compounds in mushrooms. EUC is the total amount of umami substances (umami amino acids and 5′-nucleotides) per 100 g of dry sample, usually expressed in terms of monosodium glutamate (MSG) content, also known as MSG equivalence, with units of g MSG·100 g^−1^. The higher the EUC value, the stronger the umami taste [12]. The relative odor activity value (ROAV) is another index commonly used to evaluate the key aroma compounds in mushrooms. A compound with an ROAV ≥ 1 is considered a key aroma compound, while substances with an ROAV 0.1–1 are considered to have a modifying effect on the aroma [68,69].

The summary of reports on the flavor components from multiple samples and processes of *S. rugoso-annulata* is shown in Table 4.

As shown in Figure 2, the non-volatile flavor components of *S. rugoso-annulata* mainly include taste peptides, soluble sugars, free amino acids, organic acids, 5′-nucleotides, flavonoids, alkaloids, polyphenols, inorganic salts, etc., which are water-soluble substances [12,13,14,27,32,71]. The volatile flavor compounds of *S. rugoso-annulata* include aldehydes, alcohols and ketones, esters, alkanes, alkenes, acids, ethers, phenols, heterocycles (furans, pyrazines), and other classes. Among them, aldehydes, alcohols, and ketones are the main aromatic components with higher concentrations, which all contain octanoids (eight-carbon compounds) [6,12,31,68,72].

The characteristic flavors and corresponding components of both fresh and dried *S. rugoso-annulata* mushrooms include umami taste (EUC, umami amino acids, and 5’-nucleotides), sweet taste (sweet amino acids), mushroomy aroma (octanoids), and green, grassy, and fruity aroma (aldehydes). The comprehensive flavor of *S. rugoso-annulata* mushrooms that combines both taste and aroma is VFD > HAD > ND > fresh > MWD [12]. The characteristic taste components produced by the fermentation mycelium of *S. rugoso-annulata* include glucose (sweet taste), arginine, leucine, flavonoids (bitter taste), acetic acid, citric acid (sour taste), and peptides (kokumi and rich taste) [13].

### 4.1. Taste (Non-Volatile Flavor) Components of S. rugoso-annulata

#### 4.1.1. Taste Peptides

*S. rugoso-annulata* mushrooms are a premium source of umami peptides. Free peptides in the fruiting body of *S. rugoso-annulata* reach a content of 11.28–12.56% [73]. As shown in Table 5, the taste peptides of *S. rugoso-annulata* exhibit a strong umami taste, accompanied by saltiness, bitterness, richness, or aftertaste [5,74,75]. The taste peptides in the mature fruiting body are pentapeptides to undecapeptides and the taste threshold is relatively low (0.117–0.640 mmol·L^−1^) [5,74], while the peptide variety in the mycelium is particularly rich and the taste peptides are mostly heptapeptides to decapeptides [74].

#### 4.1.2. Soluble Sugars

Soluble sugars are sweet in taste. As shown in Table 6, the fruiting body of *S. rugoso-annulata* contains various soluble sugars or polyols such as trehalose, arabinose, glucose, mannose, fructose, galactose, xylose, ribose, mannitol, erythritol, and arabitol, with a total content range of 10.01–13.06% [12,27]. Trehalose (7.82–11.03%) or arabinose (5.46–26.54%) have the highest content [12,27], indicating that *S. rugoso-annulata* mushrooms, particularly the stipes, are excellent raw materials for extracting these two sugars. The mycelium of *S. rugoso-annulata* contains glucose, gluconic acid, galactose, rhamnose, and ribose, with a total content of 8.32–8.65%. Glucose has the highest content (7.92–8.25%) in the mycelium [13].

#### 4.1.3. Free Amino Acids

In terms of free amino acids, Glu and Asp provide the umami taste while Thr, Ser, Pro, Gly, and Ala provide the sweet taste. Phe, Arg, His, Val, Met, Ile, Leu, and Trp provide the bitter taste, and other amino acids are tasteless [27]. As shown in Table 7, the fruiting body of *S. rugoso-annulata* contains 15–21 free amino acids, with a total content range of 3.07–9.38%. Thr or Glu have been reported to have the highest content [12,27,30,32]. The content of taste amino acids follows the order of sweet amino acids > bitter amino acids > umami amino acids [14,30,32]. In the fermented mycelium of *S. rugoso-annulata*, there are 20 free amino acids with a total content of 2.27–2.48%, and Arg has the highest content. The content of taste amino acids follows the order of bitter amino acids > sweet amino acids > umami amino acids [13].

#### 4.1.4. 5′-Nucleotides

5′-Nucleotides provide umami taste and can significantly reduce the perception threshold of umami amino acids at low concentrations, showing strong synergistic flavor-enhancing effects [30]. *S. rugoso-annulata* contains six 5′-nucleotides, including 5′-cytidylic acid (C), 5′-uridylic acid (U), 5′-guanylic acid (G), 5′-inosinic acid (I), 5′-xanthylic acid (X), and 5′-adenylic acid (A) (Table 8). The total content of these nucleotides ranges from 0.47% to 1.07% [12,14,27,32]. Four nucleotides, including G, I, X, and A, are referred to as umami nucleotides [27], which are also the purine component in *S. rugoso-annulata* and provide an intense umami taste.

#### 4.1.5. Organic Acids

Organic acids provide a sour taste. *S. rugoso-annulata* contains seven organic acids, including malic acid, tartaric acid, ascorbic acid, acetic acid, citric acid, fumaric acid, and succinic acid (Table 9). The total content of these organic acids ranges from 11.10 to 24.14% [13,14,27,30]. In the fruiting body, malic acid has the highest content (9.2–16.77%) [27,30], while in the fermented mycelium, acetic acid has the highest content (0.42%) [13].

#### 4.1.6. Other Taste Components

Alkaloids and flavonoids provide a bitter taste, with contents of 0.016–0.020% and 0.35–0.41% respectively, in the mycelium of *S. rugoso-annulata*. Polyphenols produce astringency, with a content of 0.0063–0.014% in the mycelium [13]. *S. rugoso-annulata* mushrooms contain a relatively high amount of K (3.41%) and Na (0.12%) [2], which also contribute to their taste. K^+^ and Na^+^ provide a salty taste [76], and Na^+^ can also combine with glutamine or succinic acid to form MSG or sodium succinate, respectively. Both of these compounds are important umami substances [77].

### 4.2. Aroma (Volatile Flavor) Components of S. rugoso-annulata

The odor of fresh *S. rugoso-annulata* mushrooms is primarily characterized by mushroomy, earthy, and grassy aromas [12,70]. During the drying process of fresh mushrooms, the earthy and grassy aroma gradually decreases, while a burnt and malty odor emerges and becomes stronger. However, the mushroomy and earthy aroma remains the key characteristic aroma of both fresh and dried *S. rugoso-annulata* mushrooms.

The characteristic aroma components profile of *S. rugoso-annulata* fruiting body consists of 19 compounds common to fresh and dried mushrooms, which mainly belong to aldehydes, alcohols, ketones, and octanoids (Figure 3) [12]. The key aromatic components that contribute the most to the aroma of *S. rugoso-annulata* are aldehydes, including isovaleraldehyde, hexanal, etc., and octanoids, including 3-octanone, 3-octanol, 1-octene-3-one, 1-octen-3-ol, etc. Aldehydes were the most abundant both in number and contents [6,12,70]. Octanoids give *S. rugoso-annulata* its mushroomy and earthy smell. 1-octen-3-ol, also known as Fungusol, is a compound found in many mushrooms and is primarily responsible for the characteristic mushroomy smell [67]. The strong earthy odor in fresh *S. rugoso-annulata* mainly results from the odor of 3-octanone and 1-octen-3-one [70]. Hexanal in aldehydes imparts a grassy aroma and isovaleraldehyde imparts a malty and fruity aroma, while heterocyclic compounds such as furans, pyrazines, and pyridines give the burnt smell during the process of heating [6,7,12,31,70].

### 4.3. Factors Influencing Flavor Components of S. rugoso-annulata

#### 4.3.1. Taste Components in Different Parts of *S. rugoso-annulata*

There was no difference in the variety of taste components between the pileus and stipe of *S. rugoso-annulata*, but there were significant differences in the content (Table 10). The pileus’s content of free amino acids, organic acids, 5′-nucleotides, and EUC were significantly higher than those in the stipe. The content of soluble sugars in the stipe was significantly higher than that in the pileus [27]. These indicate that the pileus of *S. rugoso-annulata* is more umami-rich, whereas the stipe tends to be sweeter in taste.

#### 4.3.2. Aroma Components in Different Parts of *S. rugoso-annulata*

The fresh pileus and stipe of *S. rugoso-annulata* were found to contain 15 and 14 volatile components, respectively, with 3-octanone, schisterol, and hexanal being the same major ones. 3-octanone was the compound with the highest content (>70%) in both parts. Hexanal, 3-ethyl-2-methyl-1, 3-hexadiene, and pentadecane were found in the pileus but not in the stipe, while α-bisabolene and dehydrovomifoliol were detected in the stipe but not in the pileus. The pileus had a higher content of aldehydes, esters, alkenes, and alkanes, while the stipe had a higher content of alcohol [31].

The dried pileus and stipe of *S. rugoso-annulata* (first subjected to HAD and then FVD) were found to contain a total of 50 volatile components, respectively, with alcohols, esters, and alkanes being the main constituents in the pileus, and alcohols, esters, and ketones being the main constituents in the stipe. The dried pileus exhibited a higher content of alkanes, whereas the dried stipe showed a higher content of ketones [72].

#### 4.3.3. Taste Components of *S. rugoso-annulata* at Different Developmental Stages

As shown in Figure 4, the content of free amino acids (2.48%) and 5′-nucleotides (0.0067%) in fermented mycelium is obviously lower than in the fruiting body [13], resulting in an almost negligible umami taste (EUC) in the fermented mycelium. The total content of free amino acids, 5′-nucleotides, EUC, and TAV in the fruiting body showed a trend of first increasing and then decreasing [32].

S2 showed the highest 5′-nucleotide content (0.91%), TAV (112.15), and EUC (788.66 g MSG·100 g^−1^). S4 showed the highest content of total free amino acids (9.38%), umami amino acids (1.29%), and sweet amino acids (1.70%) [32]. These findings serve as a reference for determining the optimal timing for harvesting. *S. rugoso-annulata.* This study provides a reference for the timely harvesting of *S. rugoso-annulata* rich in nutrition and taste components.

#### 4.3.4. Taste Components of *S. rugoso-annulata* with Different Processing Methods

In the preparation of *S. rugoso-annulata* soup (water extracts of fruiting body powder), processing methods present a significant influence on the content of taste components. The non-thermal treatment of HHP resulted in the highest levels of 5′-nucleotides (1.46–1.82%) and free amino acids (10.43–12.36%) and the highest EUC (827.44–1411.79 g MSG·100 g^−1^), while thermal treatments (70 °C and 90 °C heating) led to the highest concentration of organic acids (6.55–9.84%). Other non-thermal treatments, including UT and HG, led to higher levels of soluble sugars (1.56–3.05%) [14]. These findings are helpful in the cooking of highly flavorful *S. rugoso-annulata* soup.

Grinding the *S. rugoso-annulata* powder (particle size) also has a significant impact on the release of taste components into soup. Moderately ground powder (36.63 μm) leads to the highest concentration of free amino acids and soluble sugars in the soup. These findings suggest that HHP and moderately fine grinding are beneficial for the extraction of taste components from *S. rugoso-annulata* powder [14].

Different drying methods have been found to result in significant variations in the levels of taste components in *S. rugoso-annulata* [12,30,71]. HAD samples had the highest levels of total free amino acids (7.36%), sweet amino acids (2.85%), and bitter amino acids (1.91%). VFD samples had the highest contents of total organic acid (24.14%), umami amino acids (1.05%), and 5′-nucleotides (1.07%). The EUC values of different *S. rugoso-annulata* samples were ranked in the order of HAD (229.87 g MSG·100 g^−1^) > VFD (177.00 g MSG·100 g^−1^) > MWD (159.50 g MSG·100 g^−1^) [30].

HAD-dried *S. rugoso-annulata* exhibited the highest EUC value, while VFD-dried samples had the highest levels of taste components [27,30]. HAD and VFD have been found to be beneficial for preserving and enhancing the taste components in *S. rugoso-annulata*, making them suitable for the production of tasteful dried mushrooms [12,30]. Among them, HAD is the most economical and convenient drying method. On the other hand, MWD was shown to cause a significant decrease in the content of taste components and therefore is not suitable for drying.

#### 4.3.5. Aroma Components of *S. rugoso-annulata* with Different Processing Methods

A total of 47 volatile components were identified in the *S. rugoso-annulata* soup, primarily consisting of alcohols, aldehydes, ketones, esters, hydrocarbons, and a small number of heterocyclic compounds (Figure 5). The main odor components (with higher concentrations) were hexanol, 1-octen-3-ol, and hexanal. The ranking of the processing methods for the aroma intensity of *S. rugoso-annulata* soup from high to low was UT > HHP > RT > HG > 70 °C > 90 °C. The non-thermal-treated soup displayed a more intense aroma than the thermal-treated soup. Thermal treatment reduced the volatile components of the soup, and the higher the processing temperature, the more severe the loss of volatile compounds. Grinding the mushroom powder (particle size) influenced the odor of the mushroom soup. Ultrafine grinding probably destroyed the volatile components of *S. rugoso-annulata* powder. The soup of non-ultrafine-ground mushroom powder (181.25 μm) had the strongest aroma [15].

Drying significantly affects the odor of *S. rugoso-annulata*; the variety of odor compounds in dried *S. rugoso-annulata* (59–68) is significantly higher than that of fresh *S. rugoso-annulata* (47). Different drying methods result in different aroma profiles—fresh mushrooms have a slightly grassy and earthy aroma, while HAD mushrooms have a mild nutty aroma with a decrease in grassy flavor. VFD mushrooms retain a more distinct grassy flavor; MWD mushrooms have a slightly nutty, burnt, and sulfurous smell; ND mushrooms have a slight sulfur odor. Among the drying methods, VFD can best preserve the odor quality of fresh mushrooms [12].

Deep frying significantly increased the richness of the aromas of *S. rugoso-annulata* stipe, adding roasted nut and chocolate aromas. In the unfried samples, the volatile flavor compounds were mainly alcohols and esters, and 1-octene-3-one contributed the most to the aroma. In the fried samples, aldehydes were found the most, mainly including n-hexanal, octanal, nonanal, and (E)-2-nonenal, with (E)-2-nonenal making the largest contribution to the aroma [7]. This finding provided a reference for the deep-frying processing of *S. rugoso-annulata*.

#### 4.3.6. Flavor Components of *S. rugoso-annulata* at Different Temperatures

Rao et al. reported the effects of storage temperature (0 °C, 15 °C) on post-harvest quality and volatile flavor components [68]. The results revealed that fresh *S. rugoso-annulata* after harvest can be stored at 15 °C for 4–6 days and at 0 °C for 12 days while maintaining good quality without decaying. 3-octanol, 1-octene-3-ol, (E)-2-octenal, nonanal, decanal, 3-octanone, and 2-pentylfuran were the characteristic flavor substances in fresh *S. rugoso-annulata*. As the storage time extended, 3-octanol and hexanal were the key substances for the aroma deterioration of post-harvest *S. rugoso-annulata*. Compared to the samples stored at 15 °C, *S. rugoso-annulata* samples stored at 0 °C showed a lower ROVA of undesirable aroma components and a greater variety of aroma compounds [68]. This result indicated that 0 °C storage could effectively maintain the post-harvest quality of *S. rugoso-annulata* and inhibit its flavor deterioration, providing valuable references for maintaining the post-harvest storage of *S. rugoso-annulata*.

The processing temperature adopted for preparing *S. rugoso-annulata* soup led to significant differences in the content of 5’-nucleotides. When the soup is prepared at 70 °C or 90 °C, 5′-CMP and 5′-XMP are detected, unlike in soups prepared at room temperature. The total content of 5′-nucleotides in the soup follows a descending order of RT > 70 °C > 90 °C [14]. Additionally, the total concentration of volatile flavor components in *S. rugoso-annulata* soup follows a descending order of RT > 70 °C > 90 °C [15].

Qin et al. reported on the effect of various drying temperatures of HAD on the volatile flavor components of *S. rugoso-annulata* [38]. Within the range of 30–90 °C, as the HAD temperature increased, the diversity of volatile components decreased, and more alkanes were produced. A HAD temperature of 30 °C led to the most abundant volatile components (58) in *S. rugoso-annulata* [31].

Bao et al. reported the effects of roasting on the volatile flavor compounds of *S. rugoso-annulata* [6]. The result showed that roasting *S. rugoso-annulata* within the temperature range of 100–180 °C improved the richness of flavor, increased the malty, rose-like, nut-like, and cocoa-like aromas, and improved the overall flavor of *S. rugoso-annulata.* The largest number (63) of flavor components was found in *S. rugoso-annulata* roasted at 140 °C. With the increase in roasting temperature, the content of alcohols decreased significantly, while the content of aldehydes, alkanes, and pyrazines increased [6]. The results indicated that roasting enhanced the volatile flavor of *S. rugoso-annulata*.

The adsorption temperature (40–100 °C) in the microextraction step of headspace solid-phase microextraction gas chromatography–mass spectrometry (HS-SPME-GC-MS) significantly influenced the detection result of volatile components. The number of detected volatile compounds was positively correlated with the adsorption temperature, with only one for 40 °C and 18 for 100 °C. The optimal adsorption temperature was found to be 80 °C [31].

#### 4.3.7. Comparison of Aroma Components between *S. rugoso-annulata* and Other Edible Mushrooms

The aroma components of *S. rugoso-annulata* are also related to its genetic characteristics. Different edible mushroom species have their own characteristic odor components, and there can be significant differences in aroma profiles between them (Table 11). The aroma components of *S. rugoso-annulata* do not contain sulfur compounds [12,70], whereas those of shiitake (*Lentinula edodes*) and straw mushroom (*Volvariella volvacea*) include sulfur compounds as the main and characteristic aroma components [78,79]. Sulfur compounds generally release a strong, unpleasant smell [80]. Therefore, unlike shiitake or straw mushrooms, *S. rugoso-annulata* mushrooms do not have an unpleasant odor, giving a mild earthy and grassy aroma.

Generally, the flavor components of *S. rugoso-annulata* are influenced by various internal factors such as genetic characteristics, developmental stages, and parts of the mushroom, and external factors such as processing methods and temperature conditions.

## 5. Problems and Prospects

### 5.1. Zero Research Report on Chitin

As the main component of fungal cell walls, chitin is a structural polysaccharide composed of N-acetylglucosamine linked by β-1, 4 glycosidic linkages. It is widely present in fungi and crustaceans and was first discovered in mushrooms [81]. It is insoluble in water but can only be dissolved in strong acids [82]. The reason why mushrooms are not easily cooked to softness after prolonged cooking is because they contain a large amount of chitin [83].

The chitin contents of *A. bisporus*, *P. ostreatus*, *L. edodes*, *F. velutipes*, *P. eryngii*, and *V. volvacea* have been reported, ranging from 4.77% to 19.6% [84]. However, the chitin content of *S. rugoso-annulata* has not yet been reported and related studies are still lacking.

### 5.2. High Protein Content Distortion

*S. rugoso-annulata* mushrooms have a significantly high protein content (25.75–34.17%), being 1.8 times that of *A. bisporus*, 1.4 times that of *L. edodes*, and 1.3 times that of *P. ostreatus* [16,85]. However, in some studies, the protein content of *S. rugoso-annulata* has been reported to exceed 50% [32]. This could be due to the incorrect conversion factor when measuring the protein content using the Kjeldahl method.

The Kjeldahl method is a classic method for measuring the nitrogen and protein contents of samples and is included in the standard for determining protein content in food. This method determines the protein content by measuring the total nitrogen content and multiplying it by a conversion factor (nitrogen to protein factor, NPF), generally taken as 6.25 [86]. However, because edible mushrooms contain a large amount of non-protein nitrogen, in chitin, the content of non-protein nitrogen needs to be excluded when measuring the protein content of edible mushrooms using the Kjeldahl method.

To obtain a more accurate protein content, the NPF for edible mushrooms is generally modified to 4.38 [27,87]. After modifying the conversion factor, the protein content of *S. rugoso-annulata*, which was originally reported to exceed 50%, decreased to around 35%, which is a more realistic and accurate figure.

### 5.3. Little Research on the Bioactive Components in the Fermentation Mycelium and Liquid of S. rugoso-annulata

Mycelium and fermentation liquid produced through the liquid fermentation of edible mushrooms are important sources of pharmacologically active ingredients. Liquid fermentation has advantages such as a short cycle, year-round production, and controllable quality. Secondary metabolite production can also be increased intentionally by changing fermentation parameters and cultivation methods [88]. Currently, the bioactive ingredients of *S. rugoso-annulata* are mainly derived from its fruiting bodies, and there is little research on the bioactive ingredients and pharmacological activities of mycelium and fermentation liquids. Therefore, it is necessary to further conduct research on the liquid culture of *S. rugoso-annulata* mycelium and the bioactive components of its products.

### 5.4. Genes Related to the Metabolism of Nutritional and Flavoring Components Have Not Been Reported

Currently, research progress in the biosynthetic pathways and metabolic regulation mechanisms of key volatile components in *S. rugoso-annulata* has already been made. For example, the biosynthetic pathway of the main odor components (aldehydes, ketones, and alcohols) generated during the drying process of *S. rugoso-annulata* was found to be the lipoxygenase (LOX) metabolic pathway, and the key enzymes involved in regulation are lipoxygenase and alcohol dehydrogenase (ADH). LOX catalyzes the formation of aldehydes and ketones of C_6_, C_8_, and C_9_, while ADH catalyzes the reduction of aldehydes and ketones into corresponding alcohols [70], but the research findings are still in the preliminary stage. Research on nutritional and flavor components needs to delve into the genetic level in order to fully understand the mechanisms of component metabolism. So far, there have been no reports on the important functional genes of *S. rugoso-annulata*.

### 5.5. The Deep-Processed Products Are Almost Blank

In the current Chinese market, *S. rugoso-annulata* products are mainly limited to fresh mushrooms, with only a small quantity of dried and pickled mushrooms available [89]. The products are limited and quite primary. As an excellent raw material, *S. rugoso-annulata* mushroom can be used to extract various nutrients, bioactive ingredients, and flavor components.

Compared with other edible mushroom varieties, *S. rugoso-annulata* is not only rich in nutritional value such as protein and amino acids, but also abundant in bioactive and flavor substances such as polysaccharides, sterols, and taste peptides. There are great application prospects for *S. rugoso-annulata* in the fields of dietary nutrition supply, bioactive compound extraction, functional food production, flavor food processing, etc. 

## Figures and Tables

**Figure 1 jof-09-00792-f001:**
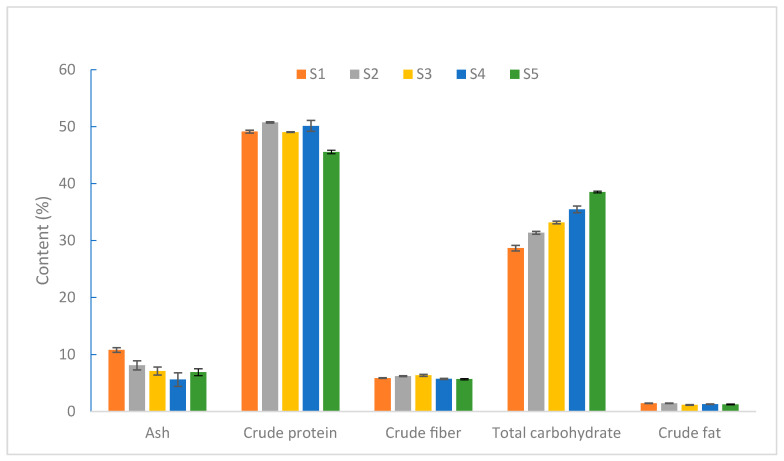
The content of macronutrients of *S. rugoso-annulata* (HAD) at different developmental stages (data obtained from reference [34]).

**Figure 2 jof-09-00792-f002:**
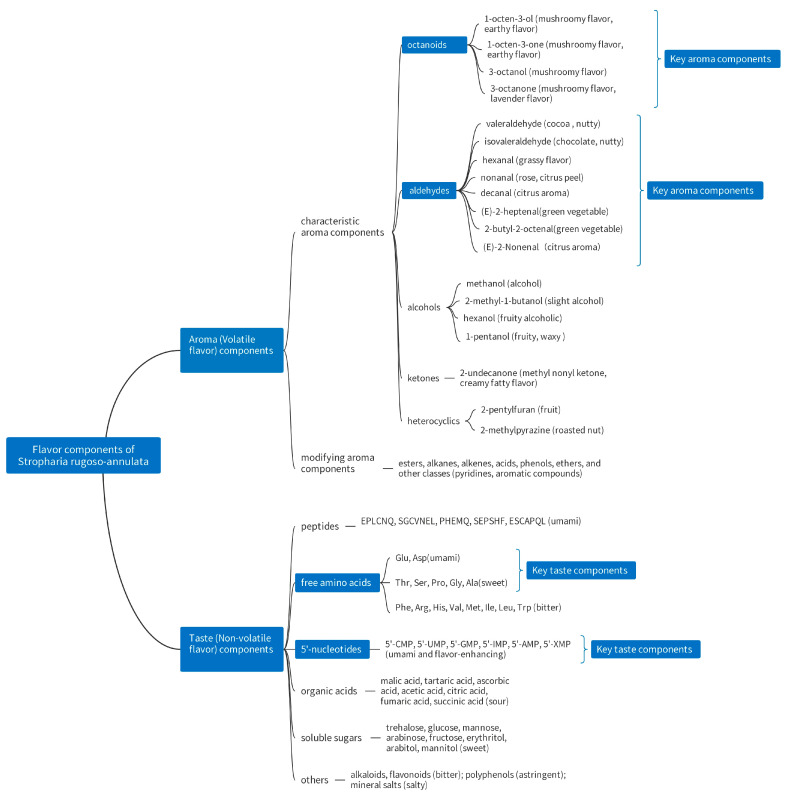
Flavor components in *S. rugoso-annulata* (data obtained from references [12,13,14,27,30,32]).

**Figure 3 jof-09-00792-f003:**
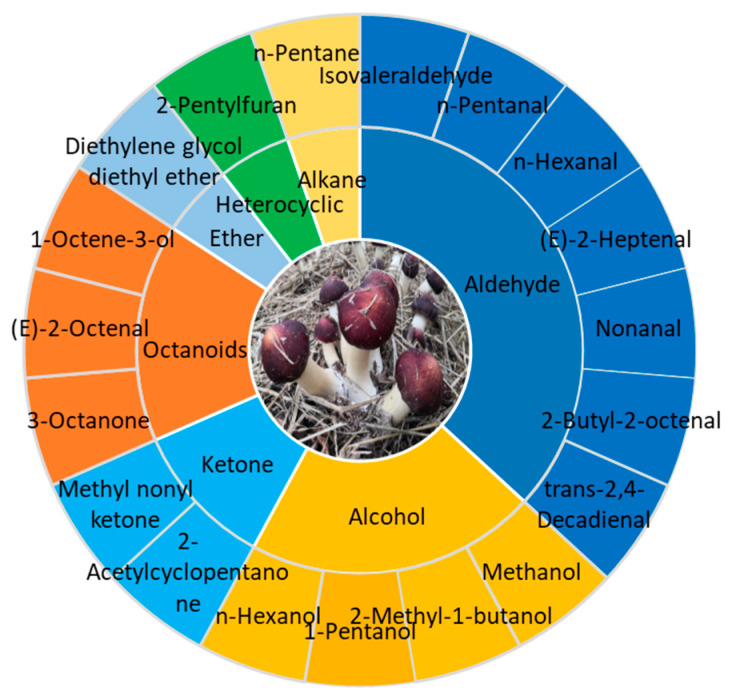
Characteristic aroma components profile of *S. rugoso-annulata* (data obtained from reference [12]).

**Figure 4 jof-09-00792-f004:**
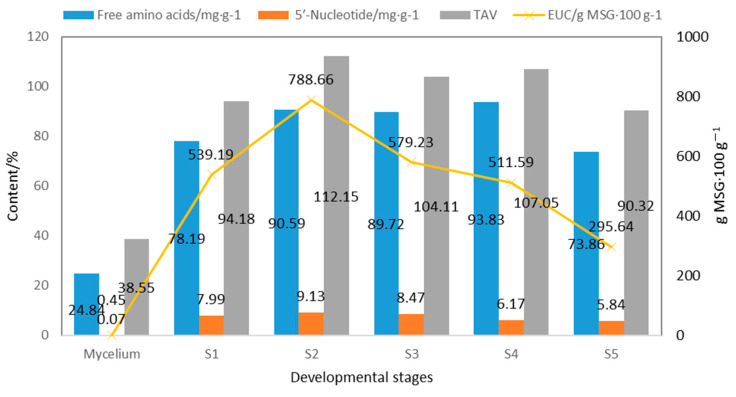
Taste components and TAV at different developmental stages of *S. rugoso-annulata* (data obtained from references [13,32]).

**Figure 5 jof-09-00792-f005:**
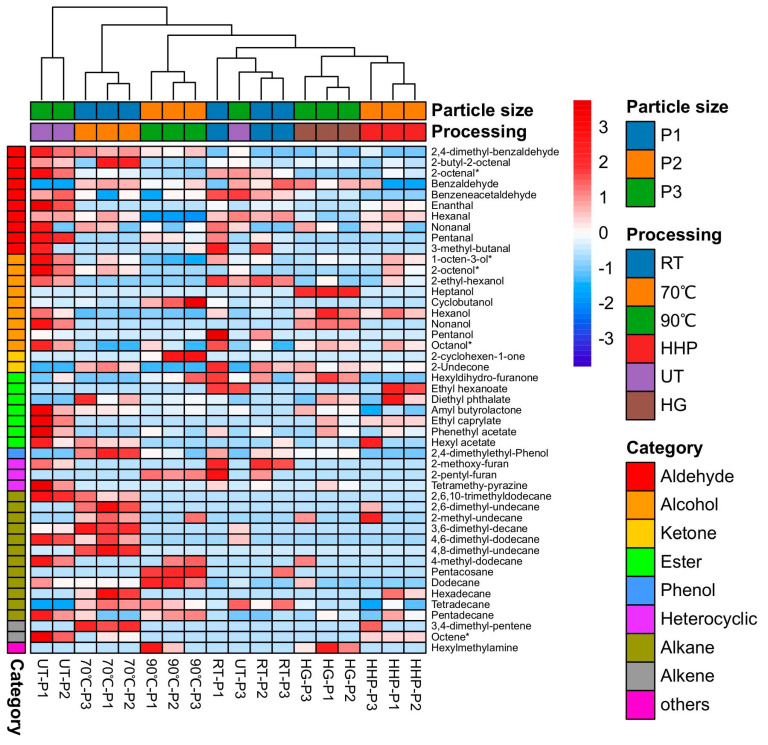
Heatmap of the volatile components of *S. rugoso-annulata* soup processed by six different methods and 3 levels of particle sizes (data obtained from reference [15]). Notes: * means octanoids. P1—particle size 181.25 μm; P2—particle size 36.63 μm; P3—particle size 7.06 μm; RT—room temperature; HHP—high hydrostatic pressure; UT—ultrasound treatment; HG—homogenization.

**Table 1 jof-09-00792-t001:** Content of nutritional components in *S. rugoso-annulata* pileus and stipe.

Parts of *S. rugoso-annulata*	Contents of Nutritional Components									
	Crude Protein (%)	Total Amino Acid (%)	Ash (%)	Crude Fat (%)	Total Carbohydrate (%)	Crude Fiber (%)	Mn (mg·kg^−1^)	Zn (mg·kg^−1^)	Ca (mg·kg^−1^)	Mg (mg·kg^−1^)
Pileus	26.07–34.60 *	17.51–28.19 *	9.95–11.36 *	2.43–3.83 *	46.24–49.10	5.05–6.97	7.3 *	91.4 *	183.3	688.7
Stipe	12.49–22.60	6.68–20.40	7.71–8.88	1.31–2.16	58.91–60.80 *	5.48–11.26 *	4.3	35.2	218.8 *	1211.3 *

Notes: Data obtained from reference [16,27,36]; * indicates significantly higher difference (*p* < 0.05).

**Table 2 jof-09-00792-t002:** Monosaccharide components, structure and optimum extraction conditions of soluble polysaccharides in *S. rugoso-annulata*.

Source	Monosaccharide Components/Molar Ratio	Structure Characteristic	Molecular Weight	Optimal Extraction Conditions	Yield	References
Fruiting body	Glucose, galactose, fucose (61.48:35.19:3.33)	The main chain was composed of [→3)-Glcp-(1→] and [→3,6)-Glcp-(1→], and the branch chain was composed of [→6)-Galp-(1→], T-Glcp and a small amount of T-Fucp exists at the end	22.907 kDa	80 °C, 3 h, 1:30 (*w*/*v*)	3.46%	[41]
Fruiting body	D-fructose, D-glucose, D-xylose, two unidentified monosaccharides	Mainly pyranose conformation, with both α- and β-glycosidic bonds	22 kDa	73.8 °C, 22.7 min, 1:26.2 (*w*/*v*)	7.64%	[53]
Fruiting body	Galactose, glucose (3:1)	A skeleton structure of (1,6)-α-D-Galp with branches at 2-O and the branches consisting of a (1→6,4)-β-D-Glcp and two →2)-α-D-Glcp	13.28 kDa	100 °C, 6 h, 1:20 (*w*/*v*)	——	[42]
Fruiting body	SRP-1 (neutral polysaccharides) consist of galactose, glucose, and mannose (4.11:56.26:1); SRP-2 (acidic polysaccharides) consist of galactose, glucose, glucuronic acid, mannose, and ribose (1.41:23.24:1.22:1:3.03)	Both neutral polysaccharides and acidic polysaccharides have a (1→6)-α-D-glucan backbone; neutral polysaccharides have both α-type and β-type glycosidic linkages whereas acidic polysaccharides have a dominant β-type constitution	——	Room temperature, microwave 640 W, 2 min, 1:50 (*w*/*v*)	——	[49]
Fruiting body	The same four types of monosaccharides: mannose, glucose, galactose, and arabinose; FSRP–1 (2.46:25.39:7.70:1); HSRP–1 (1.53:31.02:4.34:1); FSRP–2 (3.55:19.19:6.20:1); HSRP–2 (0.37:13.79:5.63:1)	Neutral polysaccharides FSRP-1 and HSRP-1 have the same (1→6)-α-D-Galp as the backbone chain, acidic polysaccharides FSRP-2 and HSRP-2 have the same (1→3)-β-D-Glcp as the backbone chain. FSRP-1 contains two types of side chains, (1→6)-β-D-Glcp and (1→3)-β-D-Glcp, HSRP-1 only contains the (1→3)-β-D-Glcp side chain. Both FSRP-2 and HSRP-2 contain the same (1→4)-α-D-Galp side chain	——	room temperature, microwave 640 W, 2 min, 1:50 (*w*/*v*)	——	[28]
Fruiting body	——	Pyranose conformation	2456–8992 kDa	100 °C, 2 h, 1:30 (*w*/*v*)	6.22–10.7%	[26]
Fruiting body	Glucose, galactose, glucuronic acid, fructose, xylose, fucose, arabinose, rhamnose (35.79%:26.80%:9.92%:8.65%:7.92%:4.19%:3.46%:3.26%)	——	27.52 kDa	100 °C, 2 h, 1:30 (*w*/*v*)	6.85%	[3]
Mycelium (intracellular)	Mannose, ribose, rhamnose, glucose, xylose, arabinose, fructose (0.76:1.64:0.65:1.00:1.24:1.18:0.20)	Pyranose conformation, β-glycosidic bonds	——	82 °C, 3.5 h, 1:26 (*w*/*v*)	15.12%	[50]
Mycelium (extracellular)	Mannose, glucose, galactose (44.4%:7.7%:1.2%)	——	——	——	——	[48]
Mycelium (extracellular)	——	β-glycosidic bonds	5.305 kDa	(Liquid medium per liter) 60.0 g sucrose, 6.0 g tryptone, 5 mM KH2PO4, and initial pH 7.0, 28 °C	9.967 g/L	[40]

Notes: —— means no data.

**Table 4 jof-09-00792-t004:** Research reports on the flavor components of *S. rugoso-annulata*.

Samples	Processing Methods	Flavor Components		References
Fresh fruiting body	——	Non-volatile	Volatile	[12]
Dried fruiting body	HAD (50 °C), VFD, MWD, ND (20 ± 4 °C)	Non-volatile	Volatile	[12]
Fresh fruiting body	Post-harvest storage (≤1 h, 0 °C)	——	Volatile	[31]
Fresh pileus and stipe	Post-harvest storage (≤1 h, 0 °C)	——	Volatile	[31]
Dried fruiting body	HAD (30 °C, 50 °C, 70 °C, 90 °C), VFD	——	Volatile	[31]
Fruiting body from fresh to dried	HAD (55 °C, 0–12 h)	——	Volatile	[70]
Dried fruiting body at different developmental stages	HAD (50 °C)	Non-volatile	——	[32]
Dried fruiting body	HAD (50 °C), VFD, MWD	Non-volatile	——	[30]
Dried pileus and stipe	NAD (20–30 °C), HAD (50 °C), VFD	Non-volatile	——	[27]
Fresh pileus and stipe	——	Non-volatile	——	[71]
Dried pileus and stipe	HAD (55 °C), VFD, MVD	Non-volatile	——	[71]
Fresh fruiting body	Post-harvest storage (0 °C, 15 °C, 12 d)	——	Volatile	[68]
Fresh stipe	Battered and deep-fried (180 °C)	——	Volatile	[7]
Dried pileus and stipe	HAD and then VFD	——	Volatile	[72]
Dried fruiting body	Roasted (100 °C,120 °C,140 °C,160 °C, 180 °C, 5 min)	——	Volatile	[6]
Soup (water extracts of fruiting body powder)	Thermal treatment (70 °C, 90 °C), non-thermal treatment (RT, UT, HG, HHP)	——	Volatile	[15]
Soup (water extracts of fruiting body powder)	Thermal treatment (70 °C, 90 °C), non-thermal treatment (RT, UT, HG, HHP)	Non-volatile	——	[14]
Fermentation mycelium	VFD	Non-volatile	——	[13]
Extracellular fluid of fermentation mycelium	——	Non-volatile	Volatile	[13]

Notes: —— means no data; RT—room temperature; UT—ultrasound treatment; HG—homogenization; HHP—high hydrostatic pressure.

**Table 5 jof-09-00792-t005:** Taste peptides in *S. rugoso-annulata*.

Samples	Number	Components	Taste Characteristics	Components and Taste Threshold/mmol·L^−1^	References
Fruiting body	5	Pentapeptide to heptapeptide	Umami	EPLCNQ (0.178), SGCVNEL (0.174), PHEMQ (0.390), SEPSHF (0.356), ESCAPQL (0.167)	[75]
Fruiting body	47	Octapeptide to undecapeptide	Umami	GQEDYDRLRPL (0.117), VVVGTPGRVF (0.403), ESPERPFL (0.254), HLYHPVPIEE (0.640)	[5]
Fermented mycelium	748	Heptapeptide to tetracosapeptide	Umami, salty	——	[74]

Notes: —— means no data.

**Table 6 jof-09-00792-t006:** Soluble sugars in *S. rugoso-annulata*.

Samples	Components	Total Content	Component of Highest Content	References
Dried fruiting body (HAD, VFD, MWD, ND)	Trehalose, glucose, mannose, fructose, mannitol, erythritol, arabitol	10.01–13.06%	Trehalose (7.82–11.03%)	[12]
Dried pileus (HAD, VFD, NAD)	Arabinose, trehalose, glucose	5.56–26.89%	Arabinose (5.46–26.54%)	[27]
Dried stipe (HAD, VFD, NAD)	Arabinose, trehalose, glucose	11.94–33.19%	Arabinose (11.90–32.94%)	[27]
Soup (water extracts of fruiting body powder)	Glucose, fucose, xylose, mannose, fructose, ribose	0.37–3.05%	Glucose (0.20–2.78%)	[14]
Fermentation mycelium (VFD)	Glucose, glucuronic acid, rhamnose, galactose, ribose	8.32–8.65%	Glucose (7.92–8.25%)	[13]
Extracellular liquid of fermentation mycelium	Glucose, glucuronic acid, rhamnose, galactose, ribose, arabinose, fucose	0.38–1.06 mg/mL	Glucose (0.34–1.02 mg/mL)	[13]

**Table 7 jof-09-00792-t007:** Free amino acids in *S. rugoso-annulata*.

Samples	Components	Total Content	Component of Highest Content	References
Dried fruiting body (HAD)	Glu, Asp, Thr, Ser, Pro, Gly, Ala, Phe, Arg, His, Val, Met, Ile, Leu, Trp, Lys, Tyr, Cys, Gln, Asn	7.39–9.38%	Glu (0.64–0.99%)	[32]
Dried fruiting body (HAD, VFD, MWD, ND)	Glu, Asp, Thr, Ser, Pro, Gly, Ala, Phe, Arg, His, Val, Met, Ile, Leu, Trp, Lys, Tyr, Cys	3.07–4.61%	Glu (0.47–0.77%)	[12]
Dried fruiting body (HAD, VFD, MWD)	Glu, Asp, Thr, Ser, Pro, Gly, Ala, Phe, Arg, His, Val, Met, Ile, Leu, Trp, Lys, Tyr, Cys, GABA, Orn, Tau	6.46–7.36%	Thr (1.55–1.90%)	[30]
Dried pileus (HAD, VFD, NAD)	Glu, Asp, Thr, Ser, Pro, Gly, Ala, Phe, Arg, His, Val, Met, Ile, Leu, Trp, Lys, Tyr, Cys, GABA, Orn	3.72–5.11%	Thr (0.73–1.94%)	[27]
Dried stipe (HAD, VFD, NAD)	Glu, Asp, Thr, Ser, Pro, Gly, Ala, Phe, Arg, His, Val, Met, Ile, Leu, Trp, Lys, Tyr, Cys, GABA, Orn	2.48–4.51%	Thr (0.67–0.89%)	[27]
Soup (water extracts of fruiting body powder)	Glu, Asp, Thr, Ser, Gly, Ala, Phe, His, Val, Ile, Leu, Trp, Lys, Tyr, Cys	4.10–12.36%	Leu (1.19–2.15%)	[14]
Fermented mycelium (VFD)	Glu, Asp, Thr, Ser, Pro, Gly, Ala, Phe, Arg, His, Val, Met, Ile, Leu, Trp, Lys, Tyr, Cys, Gln, Asn	2.27–2.48%	Arg (0.51–0.52%)	[13]
Extracellular liquid of fermented mycelium	Glu, Asp, Thr, Ser, Pro, Gly, Ala, Phe, Arg, His, Val, Met, Ile, Leu, Lys, Tyr, Cys, Gln, Asn	20.85–25.58 μg/mL	Ser (5.79–6.27 μg/mL)	[13]

**Table 8 jof-09-00792-t008:** 5′-nucleotides in *S. rugoso-annulata*.

Samples	Components	Total Content	Component of Highest Content	References
Dried fruiting body (HAD, VFD, MWD, ND)	G, I, A, X	0.47–0.63%	X (0.32%, HAD) I (0.13–0.26%, VFD, MWD, ND)	[12]
Dried fruiting body (HAD)	C, U, G, I, A, X	0.58–0.91%	C (0.16–0.24%)	[32]
Dried fruiting body (HAD, VFD, MWD)	C, U, G, I, A, X	0.88–1.07%	C (0.45–0.57%)	[30]
Dried pileus (HAD, VFD, NAD)	C, U, G, I, A	1.83–2.28%	C (0.96–1.32%)	[27]
Dried stipe (HAD, VFD, NAD)	C, U, G, I, A	0.82–1.19%	C (0.44–0.68%)	[27]
Soup (water extracts of fruiting body powder)	C, U, G, I, A, X	1.46–1.82%	A (0.16–1.45%)	[14]
Fermented mycelium (VFD)	C, U, G, I	0.006–0.0067%	U (0.0023–0.0026%)	[13]
Extracellular liquid of fermented mycelium	C, U, G, I, A	14.82 μg/mL–17.49 μg/mL	U (13.22–15.69 μg/mL)	[13]

**Table 9 jof-09-00792-t009:** Organic acids in *S. rugoso-annulata*.

Samples	Components	Content	Component of Highest Content	References
Dried fruiting body (HAD, VFD, MWD)	Malic acid, tartaric acid, ascorbic acid, acetic acid, fumaric acid, succinic acid, citric acid	17.68–24.14%	Malic acid (9.2–16.77%)	[30]
Dried pileus (HAD, VFD, NAD)	Malic acid, tartaric acid, ascorbic acid, acetic acid, fumaric acid, succinic acid	14.41–16.94%	Malic acid (5.17–5.96%)	[27]
Dried stipe (HAD, VFD, NAD)	Malic acid, tartaric acid, ascorbic acid, acetic acid, fumaric acid, succinic acid	11.10–16.26%	Malic acid (3.77–7.82%)	[27]
Soup (water extracts of fruiting body powder)	Citric acid, malic acid, tartaric acid, ascorbic acid, acetic acid, fumaric acid, succinic acid	4.03–9.84%	Citric acid (2.05–5.73%)	[14]
Fermentation mycelium (VFD)	Acetic acid, succinic acid, citric acid	0.55–0.61%	Acetic acid (0.42%)	[13]
Extracellular liquid of fermentation mycelium	Acetic acid, succinic acid, citric acid, malic acid	68.96–604.89 μg/mL	Acetic acid (14.43–495.13 μg/mL)	[13]

**Table 10 jof-09-00792-t010:** Content of taste components in *S. rugoso-annulata* pileus and stipe.

Parts of *S. rugoso-annulata*	Contents of Taste Components				
	Soluble Sugars (%)	Free Amino Acids (%)	Organic Acids (%)	5′-Nucleotides (%)	EUC (g MSG·100 g^−1^)
Pileus	5.56–26.89	3.72–5.11 *	14.41–16.94 *	1.83–2.28 *	219.37–784.05 *
Stipe	11.94–33.19 *	2.48–4.51	11.10–16.26	0.82–1.19	76.83–128.07

Notes: Data obtained from reference [27]; * indicates significantly higher difference (*p* < 0.05).

**Table 11 jof-09-00792-t011:** Comparison of aroma components in *S. rugoso-annulata* and other edible mushrooms.

Mushrooms	Common Names	Main Aroma Components	Characteristic Aroma Components	References
*S. rugoso-annulata*	Giant Stropharia, winecap mushroom	Octanoids, aldehydes, alcohols, ketones	Isobutyraldehyde, hexanal, 1-octen-3-ol, 1-octen-3-one, and 3-octanone	[6,12,70]
*L. edodes*	Shiitake	Sulfur compounds, octanoids, alcohols, ketones	Dimethyl disulfide, dimethyl trisulfide, methanethiol, 1-octen-3-one	[78]
*V. volvacea*	Straw mushroom	Octanoids, aldehydes, ketones	Isobutyraldehyde, hexanal, 1-octen-3-ol, methanethiol, 2-pentylfuran, dimethyl sulfide	[79]
*Agaricus bisporus*	Button mushroom, white mushroom	Octanoids, alcohols, ketones	1-octen-3-one, 3-octanone, 3-octanol	[78]
*Pleurotus eryngii*	King oyster mushroom	Octanoids, alcohols, ketones	1-octen-3-one, 1-octen-3-ol	[78]
*Pleurotus ostreatus*	Oyster mushroom	Octanoids, alcohols, ketones	1-octen-3-one, 1-octen-3-ol, 3-octanone	[78]
*Flammulina velutipes*	Golden needle mushrooms	Octanoids, alcohols, ketones	1-octen-3-one, 3-octanone, 3-octanol	[78]

## Data Availability

Not applicable.

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
