# Peer review of "Nutritional, Bioactive, and Flavor Components of Giant Stropharia (Stropharia rugoso-annulata): A Review"

_jof, 2023, doi:10.3390/jof9080792_

Round 1

Reviewer 1 Report

I congratulate the author for reviewing the about Stropharia. It is one of the important mushroom and requires the attention with respect to its edible properties.

I suggest if authors can incorporate the production data of this mushroom. 

Author Response

Point 1: I suggest if authors can incorporate the production data of this mushroom. 

Response 1: Thank you for your valuable review comments. The production data of S. rugoso-annulata has been incorporated in the second paragraph of the revised manuscript.

Reviewer 2 Report

The review is interesting but some points should be considered;

1. The important characteristic of this mushroom should be described to enhance the reader to get interest in this mushroom.

2. All data are reported only for this mushroom, it will be more interesting to compare with other mushrooms.

3. Organization/structure of this manuscript should be revised for easy reading, too many heading/sub heading.

4.  English should be edited.

5. In Fig 1, S1,. S2,... stand for ? It should be explained clearly in the legend.

6. References are mainly from China, please look for other sources too.

7. Please clearly  deliver the idea of the review.  

English should be edited.

Author Response

Point 1: The important characteristic of this mushroom should be described to enhance the reader to get interest in this mushroom.

Response 1: Thank you for your valuable review comments, and I have made the necessary modifications. The important characteristics of S. rugoso-annulata include high protein content, abundant bioactive compounds, a delicious and sweet taste, and a pleasant aroma. The specific information is described in detail in the relevant paragraphs of the revised manuscript, which I believe will spark readers' interest in it.

Point 2: All data are reported only for this mushroom, it will be more interesting to compare with other mushrooms.

Response 2: This comment is very reasonable and the comparison between S. rugoso-annulata and other edible mushrooms is also our focus for the future research work. In the revised manuscript, I have added comparisons of the nutritional components and characteristic aroma compounds between S. rugoso-annulata and other mushrooms. Due to the limitations of article length, I haven't added too much.

Point 3: Organization/structure of this manuscript should be revised for easy reading, too many heading/sub heading.

Response 3: The manuscript indeed does have the issues you mentioned. The four-level headings in the manuscript have been simplified to three-level headings, and some of the redundant text has also been removed.

Point 4: English should be edited.

Response 4: Regarding the English language revisions, I have meticulously checked and edited the manuscript three times myself and have also enlisted the thorough assistance of a foreign colleague. I believe that the revised manuscript contains no noticeable English grammar errors and conforms to the standards of English expression.

Point 5: In Fig 1, S1,. S2,... stand for ? It should be explained clearly in the legend.

Response 5: S1~S5 represent five different stages of S. rugoso-annulata fruiting body’s development process, in chronological order. This explanation has already been made in the preceding text of Fig 1.

Point 6: References are mainly from China, please look for other sources too.

Response 6: Indeed, most of the references come from China, as the S. rugoso-annulata mushroom has been very popular there in recent years and has become a hot topic in edible fungi research, resulting in abundant research findings. The research reports on S. rugoso-annulata from other countries mainly focus on cultivation and environmental pollutant degradation, with relatively fewer literature related to the components. I have already added a few new related references from other country.

Point 7: Please clearly deliver the idea of the review.  

Response 7: The purpose of this review is to provide a comprehensive and systematic summary of the research on the components, specifically the flavor components, of S. rugoso-annulata in recent years. This review aims to fill a knowledge gap by being the first of its kind in this area. By examining the influencing factors of these flavor components, this review can also offer valuable scientific evidence for the S. rugoso-annulata industry. This evidence can be utilized in the production of high-quality mushrooms, extraction of bioactive ingredients, post-harvest storage, as well as culinary processing. This, in turn, promotes the consumption of S. rugoso-annulata and contributes to the health of consumers.

Reviewer 3 Report

The document is interesting and makes a significant contribution to knowledge. However, I suggest summarizing some of the graphs if they are already described in the text instead of including the images. Additionally, it would be helpful to present the methodology in the form of diagrams for better clarity.

It is suggested to perform a stylistic and grammatical correction.

Author Response

Point 1: I suggest summarizing some of the graphs if they are already described in the text instead of including the images. Additionally, it would be helpful to present the methodology in the form of diagrams for better clarity.

Response 1: Thank you for your valuable review comments. I have removed the overlapping parts between the text and the images in the revised manuscript, as well as replaced some textual content with diagrams.

Point 2: It is suggested to perform a stylistic and grammatical correction.

Response 2: This suggestion is very necessary. Regarding the English language revisions, I have meticulously checked and edited the manuscript three times myself and have also enlisted the thorough assistance of a foreign colleague. I believe that the revised manuscript contains no noticeable English grammar errors and conforms to the standards of English expression.

Reviewer 4 Report

Dear Editor,

I am here with submitting of the review manuscript number jof-2495231 entitled " Nutritional, bioactive and flavor components of king Stropharia mushroom (Stropharia rugoso-annulata): a review".  

The authors Huang et al. presented review of literature about nutritional, bioactive, with particular reference to the aroma and taste and defining which components lead to specificity in different parts of the fruiting body of this edible mushroom, also when using different processing methods.

My suggestion is to make minor changes

-        Lines 32-34 – name some of many reports……

-       in the whole text there are some different citations of references (follow the instructions for citation);

In my opinion, the manuscript is well written, the data presented in this are interesting, and my suggestion is to accept it for publication in the Journal of Fungi

Author Response

Point 1: Lines 32-34 – name some of many reports……

Response 1: Thank you for your valuable review comments. I have added examples of four research reports in the revised manuscript, mentioning specific researchers' names and their research findings.

Point 2: in the whole text there are some different citations of references (follow the instructions for citation)

Response 2: I have made the necessary modifications to the citation format of the references in the entire manuscript, following the instructions provided by the Journal of Fungi.

Reviewer 5 Report

See in attachment.

Author Response

Point 1: it needs much careful editing.

Response 1: Thank you for your valuable review comments. Following your example, I have corrected carefully similar minor errors throughout the entire manuscript.

Round 2

Reviewer 2 Report

The manuscript is significantly improved. It could be accepted.

English was improved.